# Comparison of Structural Performance and Environmental Impact of Epoxy Composites Modified by Glass and Flax Fabrics

**Georgios Koronis \*, Arlindo Silva** and **Michael Ong**

Engineering Product Development Pillar, Singapore University of Technology and Design, Singapore 487372, Singapore
\* Correspondence: gkoronis@gmail.com

**Abstract:** Comparing the structural performance and environmental impact of parts made of natural and synthetic fibers has become increasingly important for industry and education, as the benefits of one type of fiber over another are not always clear. The current work discusses the advantages and disadvantages of using natural and synthetic fibers and compares the flexural performance of parts made of each of these fibers and their environmental impact. This paper investigates the flexural behavior of epoxy composites modified by glass and flax fabrics through experimental, numerical, and analytical studies. Specimens with various fabrics (dried and non-dried) were fabricated to test their performance. The failure of unidirectional glass and flax fiber reinforced polymer composite laminate was examined by destructive testing. A finite-element model was developed, and the mechanical behaviors of fiber-reinforced composites were predicted in a three-point bending test. Experimental results were compared to numerical analysis to validate the model's accuracy. A life cycle assessment (LCA) was employed to determine the climate impact of composite production. The analysis revealed a decreased environmental effect of plant-based panels suggesting that they are less energy and $CO_2$ intensive than synthetic solutions. The LCA model can be applied in further studies of products that consist of or use flax-based composites.

**Keywords:** green composites; environmental impact; life-cycle assessment; natural fiber reinforced polymers; experimental testing; numerical analysis

## 1. Introduction

Composite materials have been used for millennia, but the recent development of biocomposites, like most of their commercial applications, began in the late 1980s. Since then, green composites have permeated various consumer industries, including automotive, packaging, construction, and many more [1–3]. Composite materials are complex to analyze and show some performance variability depending, among other factors, on their manufacturing parameters [4]. However, there is room for decision-making in a design space composed of structural performance and environmental impact for composites with natural fibers to compete with composites made with synthetic fibers [5].

The problem of variability becomes more complex with the inclusion of natural fibers substituting for synthetic ones, even if the resin material remains the same [6]. Nonetheless, it is still possible to estimate structural performance, environmental impact, and cost at an early stage of the design of composite structures [7,8]. Natural fibers possess advantages regardless of performance variability, especially in damping noise and vibration of composite panels [9].

Forests and plant-based products can be essential in combating climate change and transitioning to a bio-based economy. Technologies have been developed that exploit the cellulose and hemicellulose from forest biomass to make fuels, chemicals, and bio-based products [10]. Recently, there has been a growing interest in using natural plant fibers such

as jute, flax, hemp, sisal, and ramie to replace fiberglass in reinforced composites [11–13]. Flax fiber has been reported to have better vibration absorption properties than other natural fibers and is less expensive and easier to obtain [14]. Furthermore, flax is stronger, sharper, and stiffer [15], making them more popular as reinforcement in polymer composites.

Many studies suggest the potential of natural fibers, particularly for lightweight construction, due to their reduced density (about 1.5 g/cm$^3$ versus 2.5 g/cm$^3$ for glass) and good mechanical properties [16–18]. One study, for example, shows that using flax fibers as reinforcing components has a lower environmental impact than using glass fibers [19]. Comparisons of cost and environmental impacts for real manufacturing scenarios with synthetic and natural fibers are also possible [20]. Further LCA studies on biocomposites conclude that adding natural fibers to replace all or part of synthetic fibers demonstrates a lower environmental impact on that component [12,21,22].

Several efforts have been made to identify the most important manufacturing parameters of composites with natural fibers allowing the development of models describing the dependence of composite resins on composition and then selecting the optimal epoxy resin compositions. However, results have been contradicting at times [6,23]. Moreover, while a direct comparison between synthetic and natural fiber composites is possible (via a modified rule of mixtures) [24], it is sometimes hard to conduct due to fundamental geometry differences (volume of fibers, the density of preforms, adaptability of/to the particular manufacturing process, etc.) [25].

Composites using renewable elements are one option to reconcile sustainable materials use and manufacturing costs in automotive panels. Plant-based solutions for automobile parts, such as trim parts in dashboards, door panels, parcel shelves, seat cushions, backrests, cabin linings, and so on, have already been examined by several automakers and academic research [26–30]. Natural fiber fabrics, as opposed to glass fabrics, provide weight reductions and, as a result, cost savings, as has been noticed in several applications. When considering updating the manufacturing process and optimizing labor, an extra 10% decrease in production costs can be realized [20].

This study uses finite-element analysis (FEA) to investigate stress interactions in unidirectional, six-ply samples loaded through a three-point bending. The mesoscale modeling of biaxial fiber reinforced epoxy composites were used to estimate the flexural properties with reasonable accuracy. A composite production was carried out in the form of laminates via a vacuum infusion process. Several similar parts made of natural or synthetic fibers with the same epoxy matrix were tested in this work to attain their mechanical and structural performance. The numerical simulation results were compared with the experimental data obtained from the destructive testing. An additional goal was to perform an environmental assessment to select materials that minimize the energy spent and $CO_2$ emissions, as green composites have the potential for many improvements in economic/mechanical performance, environmental effect, and public acceptance. The current study used the LCA tool, an established instrument for estimating the environmental impacts over their lifetime [31], where input and waste output data are collected [32].

The goals of this analysis were to (a) improve the bio-based composite production process from an environmental life cycle standpoint; (b) compare the environmental impact of the glass fiber production process to that of flax production; (c) help guide future flax production technology development by identifying environmental hotspots.

These well-established approaches (FEA and LCA) are employed as a practical way to examine an early-stage green composite's mechanical performance and environmental impact during the product's conceptual phase. In addition, these tools can guide design selections when materials and processes are not yet determined, but an approximate bending performance and energy estimate are required. Finally, it is anticipated that our findings may prove helpful to researchers (both academic and industrial), technology developers, and industry decision-makers in evaluating potential avenues for research and development of composite flax systems.

## 2. Materials and Methods

### 2.1. Standard Test Method for Flexural Properties

Flexural properties determined by D 790 are beneficial for quality control in composites [33] and may find design applications [34]. The composite samples were prepared according to the test methods for three-point flexural tests (procedure A in Figure 1), which employs a strain rate of 0.01 mm/mm/min. Flexural testing requires a simple rectangular sample in a three-point bending setup to assess specimen flexural response. Figure 1 shows how ASTM-recommended dimensions were used. The results acquisition was performed according to the ASTM D790—Standard Test Method for Flexural Properties of Polymer Matrix Composite Materials [33]. The ASTM D790 test for composites is a low-cost and simple test for determining the flexural properties of polymer-reinforced composites.

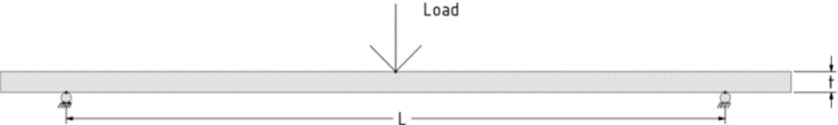

**Figure 1.** Three-point bending diagram.

The stacking sequence applied is $[(0/90)_2, (45/45)_2, (0/90)_2]$ produced by resin infusion. Six plies were chosen across systems while maintaining a variable thickness depending on the fabrics used. A diagram of the test simulated is shown in Figure 1. All coupons were assigned with a displacement of 3.5 mm as in previous studies [6,23] in analogous bending test settings.

### 2.2. Finite Element Model Geometry

Three-point bend test of composite coupons was conducted using ANSYS FEM simulation. Two metallic rollers were modeled to account for the effects of contact stresses under the loading nose. Scanning electron microscopy (SEM) was used to examine the morphology of the fabrics. In our micro-scale model, fibers were arranged hexagonally to account for high-fiber volume fractions in woven composite fiber bundles. In addition, the yarn cross-section was modeled as a lenticular shape for flax and an elliptical shape for glass variants, respectively. Table 1 lists the respective fiber volume fraction $(V_f)$ of the unit cell per variant used for the numerical analysis. Unfortunately, several values were not provided from the supplier's data sheets, and we had to draw information from online databases and, in some cases, assume missing values from earlier studies. These include the yarn fiber volume fractions and fiber diameter values. The thickness of the coupons was estimated to be simply the sum of ply thicknesses.

**Table 1.** Parameters used in the finite-element model.

| Coupon Label | Fabric Thickness (mm) | Coupon Thickness (mm) | Width (mm) | Fiber Diameter (mm) | Yarn Spacing Micro (μm) | Fiber Volume Fraction (%) |
|---|---|---|---|---|---|---|
| Plain Weave Glass | 0.18 | 1.08 | 25.4 | 10 | 130.00 | 38.8% |
| Twill Glass | 0.25 | 1.50 | 25.4 | 15 | 150.00 | 40.2% |
| Flax 200 | 0.60 | 3.60 | 25.4 | 40 | 110.00 | 30.8% |
| Flax 200 Dry | 0.50 | 3.00 | 25.4 | 40 | 110.00 | 29.4% |
| Flax 300 | 0.80 | 4.80 | 25.4 | 40 | 110.00 | 34.6% |
| Flax 300 Dry | 0.75 | 4.50 | 25.4 | 40 | 110.00 | 34.7% |

### 2.3. Meshing

Equal-size elements are applied to the base rollers and the sample to improve contact (i.e., both have a 1 mm element size). More elements were employed to capture the stress

interactions with acceptable precision towards the loading nose, where failure is expected (i.e., the load roller has a 0.5 mm element size, and the loading area on the specimen where loading is applied also has 0.5 mm size), with finer mesh near the loading and coarser away from the loading. Support rollers had a 1 mm element size, similar to the coarser part of the coupon. Figure 2 depicts the resulting mesh with three rollers and the coupon set.

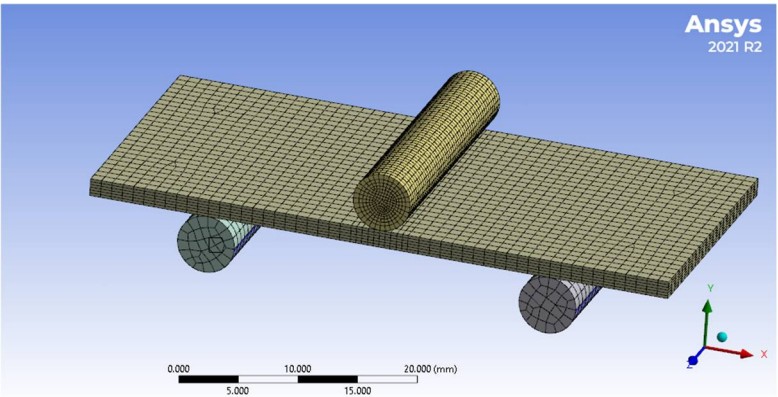

**Figure 2.** FEA mesh of coupon with rollers and loading nose.

### 2.4. Boundary Conditions and Contact Settings

Contacts were created between the sample and the loading and roller supports. The support rollers were assigned frictionless contacts, whereas the loading roller and the coupon were assigned a no separation contact. Both used a friction coefficient of zero, but the frictionless support was deemed nonlinear because the contact area may change when the load is applied. Therefore, a zero coefficient of friction is assumed, allowing for free sliding. While the sample is being loaded, some slippage may occur. However, a proper test requires no gap between the roller and the sample on the loading nose.

Three independent boundary conditions (BCs) in the form of displacements were specified to obtain the material properties of the coupons in tension. The BCs have been assigned as listed in Figure 3, and each ply's orientation was defined using a local coordinate system. The rollers were represented by solid cylinders with a radius of 2.5 mm and isotropic material properties. In the simulation, the rollers could not rotate; instead, the coupon could slide over them as it would in the lab testing machines.

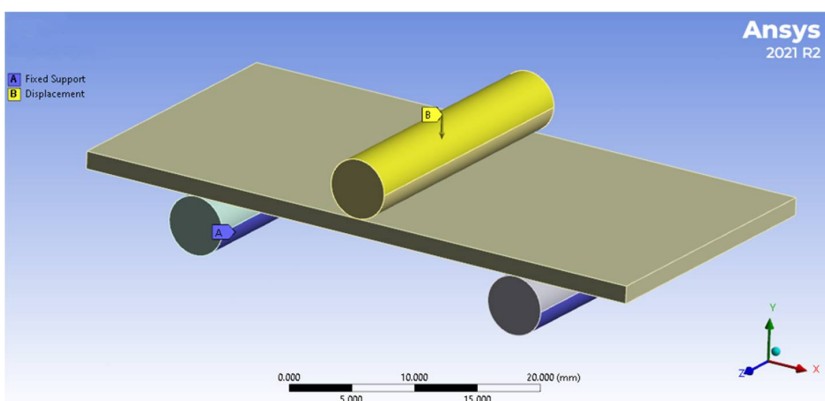

**Figure 3.** The model's contact boundary conditions.

Frictionless support restricts motion normal to the plane while permitting motion in the two other primary directions (e.g., at the XZ plane's surface, movement in the Y and rotations around X and Z are constrained). Fixed support was attached at the bottom surfaces of the support rollers, limiting all degrees of freedom at the rollers. Finally, a displacement was used at the loading roller's top. All degrees of freedom for the rollers

were limited in the same way as the fixed support was, with the addition of an enforced nodal displacement in the negative Z direction.

### 2.5. Fabrication of Bending Test Specimens

Ecotechnlin (France) supplied FlaxPLY BL550 woven textiles in balanced twill 2/2 at 550 g/m$^2$ and Flaxdry BL 200, BL 300, and Flaxply BL 200, BL 300. Wee Tee Tong Chemicals (Singapore) supplied glass fiber fabrics of 2 × 2 twill and plain weave mats made in Korea. For the matrix material, Wee Tee Tong supplied a two-part Epoxy resin EPICOTE 2820 (base resin and activator hardener) for composite fabrication. EPICOTE 2820 consists of Epicote 2820 Part A resin and Epicote 2820 Activator Part B (hardener). The properties taken from technical data sheets are given for resin in Table 2 and fabrics in Table 3.

**Table 2.** Matrix properties.

| EPOXY RESIN 2820 | |
|---|---|
| Combined Viscosity (centipoise) | 737–1000 |
| Gel Time (at 20 °C): mins | 67 min at 150 g mass |
| Flexural Strength (Mpa) | 100–110 |

**Table 3.** Fabric properties.

| Reinforcement | Weave Style | Fabrics Thickness (mm) | Density (kg/m$^2$) |
|---|---|---|---|
| Plain Weave Glass | Plain Weaving | 0.18 | 2540 |
| Twill Glass | Twill 2/2 | 0.25 | 2540 |
| Flax 200 | Twill 2/2 | 0.60 | 1270 |
| Flax 200 Dry | Twill 2/2 | 0.50 | 1270 |
| Flax 300 | Twill 2/2 | 0.80 | 1290 |
| Flax 300 Dry | Twill 2/2 | 0.75 | 1290 |

All stacking materials were put over an open mold surface (or flat glossy surface) and sealed in a bag film during the vacuum infusion process (Figure 4). Vacuum infusion is comparatively affordable and straightforward to conduct, compared to several industrial forming methods for fiber-reinforced polymers, even in a non-industrial context.

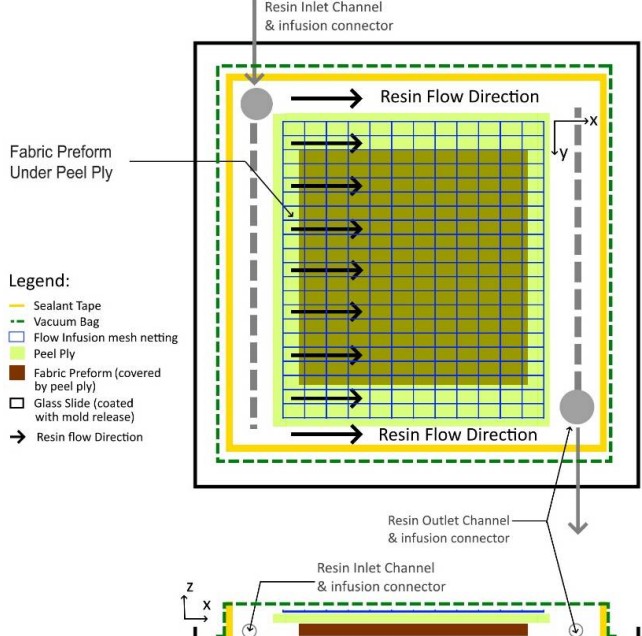

**Figure 4.** Vacuum infusion schematic (not drawn to scale).

The epoxy resin was thoroughly mixed before Infusing the stacks of fiber mats using a vacuum pump. Extra care was taken during the resin preparation to reduce the influence of variability in the plates. To reduce voids and maintain the uniformity of the composite laminas, the two-part epoxy mixture (resin and hardener) was left to settle for 10–15 min before being infused to eliminate as many air bubbles as possible. During the process, a vacuum pump applies atmospheric pressure to the laminate, and the remaining air is evacuated before the resin system is drawn in. The applied pressure serves as the mold clamp (as in traditional injection molds) and evenly distributes pressure on the laminate surface and the resin.

The bag is removed when the resin has completely cured, and the component is de-molded from the mold surface. Composite samples were cut out of these panels following the dimensions suggested by ASTM D 790 [33] for three-point bending tests. Bi-directional symmetric rectangular panels with variable nominal thickness were created at room temperature. The rectangularly shaped coupons' dimensions were 50.8 mm in length and 25.4 mm in width. Table 4 summarizes the panels' physical properties. It is important to note that the thickness used for FEM analysis (in Table 1) was an estimation, whereas the real thickness of the coupons was measured directly (in Table 4). Interestingly, they are roughly identical for natural fibers but are substantially different for glass fibers (with estimations being substantially lower than real values).

**Table 4.** Physical properties of composite panels.

| Panel Label | Resin Weight (g) | Thickness (mm) |
|---|---|---|
| Plain Weave Glass | 71 | 1.40 |
| Twill Glass | 90 | 1.85 |
| Flax 200 | 202 | 3.50 |
| Flax 200 Dry | 199 | 3.00 |
| Flax 300 | 246 | 4.80 |
| Flax 300 Dry | 242 | 4.60 |

### 2.6. Experimental Testing

Mechanical testing was performed using a Universal Instron® Flexure Testing Machine (5940 Series—single column tabletop system) outfitted with 1 KN max load pneumatic grips and a 5 mm/min crosshead speed with a 30 mm support span. The average room temperature was roughly 25 °C, with a relative humidity of about 55%. The results of the tests were summarized using computer-aided software for static systems Bluehill®3 Universal. The ultimate flexural values were obtained on the stress/strain curve's highest point following the manifestation of a failure experienced by the test coupons during testing.

### 2.7. Environmental Performance Analysis

The composite's environmental impact was quantified using the open Life Cycle Assessment (open LCA) database. The software version 1.7 was used to model the complete various production systems, compile the acquired data, and calculate the environmental impacts of the different permutations of composite plate analysis.

2.7.1. Functional Unit and Boundary Conditions for the Coupons

In this study, we assigned the composite panels as a functional unit (FU), which has an area of 420 × 190 mm. For the boundary conditions (BC), the baseline scenario considers the operational conditions of a university lab facility. The panels, the FU, have been produced using six layers of glass fiber mats in an epoxy matrix. The finished part is of 1.5 mm thickness and weighs around 168 g, out of which 97 g are glass fabrics of 37% fiber volume fraction ($V_f$%).

Figure 5 depicts the schematic diagram within the boundaries of a hypothetical life-cycle scenario regarding the FU. Energy and materials are consumed in each of the five phases of life, generating gaseous emissions outlined by the arrow shape at the bottom side

of the diagram. The system boundaries include epoxy manufacturing, fiber cultivation, textile weaving, transit, and incineration of the panels after use.

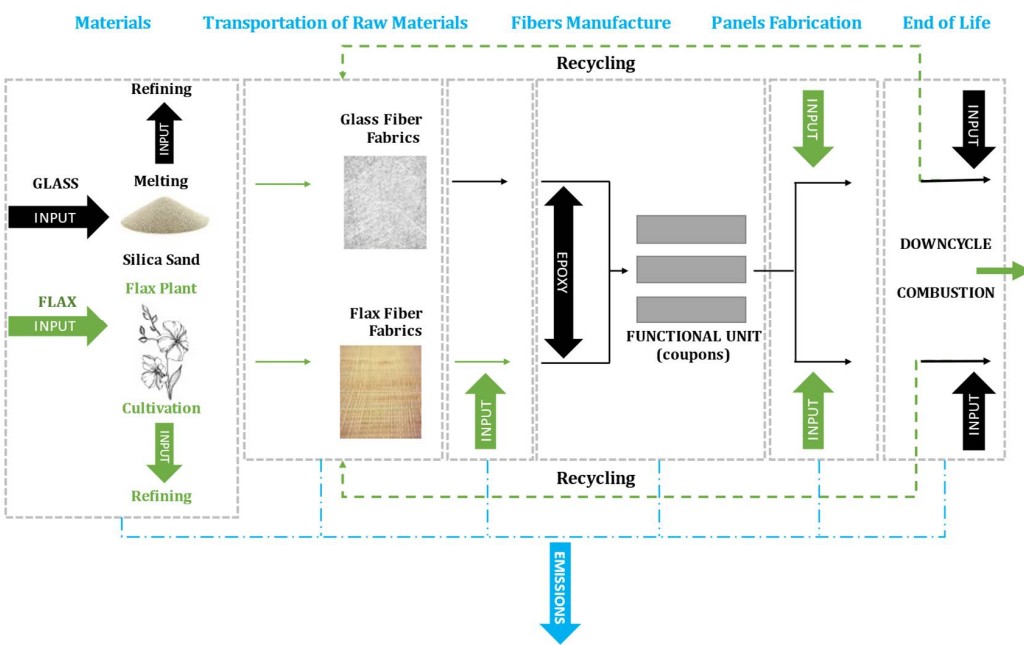

**Figure 5.** Boundaries of the eco-impact analysis for the laminate's fabrication.

### 2.7.2. Materials, Manufacture, End of Life

In general, the recyclability of bio-based plastics is extremely difficult since the infrastructure required is not generally available, and the addition of fibers may prohibit the composite from being recycled, as is the case with synthetic composites reinforced with glass fibers [35]. Furthermore, because recycling damages the properties of its constituents, incineration is a common end-of-life alternative for these materials [36]. As such, energy recovery was assumed in small percentages where the disposal phase inputs for glass scenarios and the reference composite scenario are listed according to our previous studies [20,37]. As such, mechanical recycling of composite materials can be achieved by considering (11% to 12%) due to the low recycling efficiency of bi-axial reinforced composites. The remaining percentage of granulated fractions and sieving will end up in landfills when the composite materials are disposed of.

### 2.7.3. Environmental Impact Categories

The CML-IA baseline impact technique was used to analyze the life cycle impact. This method was developed in 2001 by scholars at the University of Leiden in the Netherlands [38] and included over 1700 flows on their website. CML restricts quantitative modeling to early stages in the cause-effect chain to limit uncertainties and is divided into baseline and non-baseline categories, with the baseline being the most utilized impact category in LCA. This is a midway assessment approach based on ISO LCA standards (ISO Standard 14040), along with some modifications and upgrades [39]. To evaluate the manufacturing systems, the following impact categories were chosen from those specified in the CML approach.

- Global warming Potential (GWP). One of the primary aims of replacing fossil-based glass fiber with flax is to reduce environmental impact. The GWP is expressed in terms of fossil carbon dioxide equivalents ($CO_{2,eq}$).
- Acidification Potential (AP). Burning biomass and fossil fuels can enhance acidity owing to $SO_2$, $NH_3$, and $NO_x$ emissions. The amount of AP is measured in kg of sulfur dioxide equivalents ($SO_{2,eq}$)

- Photochemical ozone creation potential (POCP). The impact category is heavily influenced by the levels of carbon monoxide (CO), sulfur dioxide ($SO_2$), nitrogen oxide (NO), ammonium, and NMVOC (non-methane volatile organic compounds). POCP is measured in kilograms of ethylene equivalents (Kg ethylene equivalent)
- Eutrophication potential (EP). Fertilizers may be employed depending on forest management, leading to increased eutrophication. This causes excessive plant growth, such as algae, in rivers, resulting in significant reductions in water quality and animal populations. EP is measured in kg of phosphate equivalents ($PO_{4,eq}$).
- Human toxicity potential (HTP). The manufacture of glass fiber may influence human health. However, it should be emphasized that the CML technique lacks a characterization factor for hydrogen cyanide (HCN). HTP is expressed in kilograms of 1,4 dichlorobenzene equivalents ($1,4\text{-}DCB_{eq}$).

Some impact categories groups were omitted due to the incomplete list of characterization factors for our case study.

## 3. Results and Discussion

### 3.1. Analysis of Experimental Test Results and FEA Flexural Properties

In this section, we analyze the flexural behavior of the modified epoxy composites, and we compare test results obtained by destructive tests with the numerical analysis. In terms of flexural properties, the flexural strength (σf) and strain (εf), as well as the modulus of elasticity in bending (EB), were considered. Table 4 lists the destructive testing results, presenting the mean values and standard deviation for the various coupons and FEA strengths for all six systems considered in this study. According to the standards, five specimens are required to validate the properties. Plots in Figure 6 show the specimens' stress/strain curves of the five successful tests. Numerical simulations were carried out for the composite specimens, and the results are detailed next to the destructive test results in Table 4.

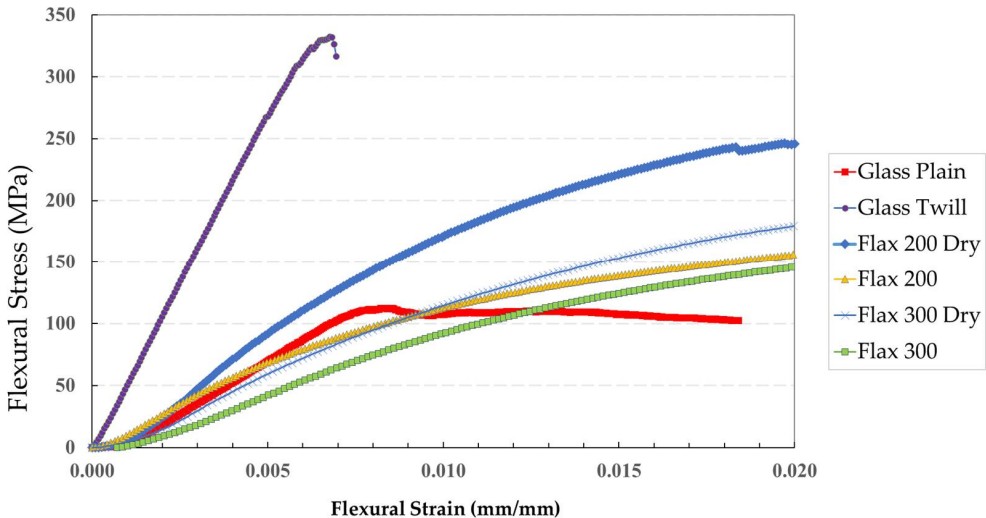

**Figure 6.** Plots of flexural strength in produced glass plain weave coupons.

There was a significant disparity between numerical simulation and experimental tests, which can be explained partially by the differences between estimated and real coupon thickness (already mentioned). Another potential source of error may be the uneven distribution of the fibers across the thickness of the coupons. This also gives the reader an idea of the errors in estimating values for initial numerical simulation screening, which then must be checked with experimental testing using the selected manufacturing process. As the process is repeated, estimations will become closer to reality, and numerically predicted values will become closer to real experimental values. The discussion that follows will focus on the experimental values only.

Results show that twill glass is the strongest specimen from the synthetic coupons, and flax 200 dry is the strongest from the plant-reinforced composites. It is also noticeable that the dried fabrics contribute positively to increasing the strength and stiffness of the variants as opposed to the wet variants. However, the lowest flexural behavior for strength and stiffness overall was noticed for the flax 300 variant. In contrast, its strength equals flax 200, which suggests that non-dried fabrics cannot achieve high performance in this six-ply setting. The lower flexural performance can also be attributed to the higher fiber loading of 30%$V_f$, of some reinforced panels, which leads to stress concentration and dispersion problems [40–42].

Unfortunately, the finite element model underpredicted flexural strength in half of the systems, and in the case of the twill glass, the strength differed significantly from the actual result. The same pattern was observed for the flexural stiffness predictions, which exhibited a great disparity with expected values, as nearly half of the prediction samples yielded 25 percentage points lower results.

We assume this occurred because those glass variants were assigned a much lower thickness than one of the produced composites, as it was 23% thicker than the one for the FEA simulation. An explanation for this could be the inconsistency of the coupon thickness that was not assumed for FEA.

### 3.2. Environmental Impacts of the Fiber Reinforced Composites

The Ecoinvent database in OpenLCA includes data for producing glass fiber in Asia and flax in Europe for the impact categories explained in Section 2.7.3. In the first level, the production phase of a glass plain weave panel is compared to flax 200 wet and flax 200 dry panels. The LCA analysis was employed for all composite systems, and we present all charts here. Still, we will be selectively focusing on specific comparisons for ease of reading and a shake of simplicity.

According to the GWP indicator, the fabrication of glass fiber reinforced panels has a climate impact of 2.79 kg $CO_{2,eq}$/kg. This impact alone is approximately 35% higher than the climate impact of flax 200 wet and 28% lower than the one of flax 200 dry (see Figure 7 and Table 5). The main contributors to GWP are fiber production (75%), resin production (12%), and air freight transport (10%). Production has the highest impact, likely due to the greater fossil-based energy associated with glass fiber manufacturing (glass melting, refining, and federalization). However, harvesting and forming flax into fiber mats is a process that is less energy-intensive compared to the emissions associated with e-glass production.

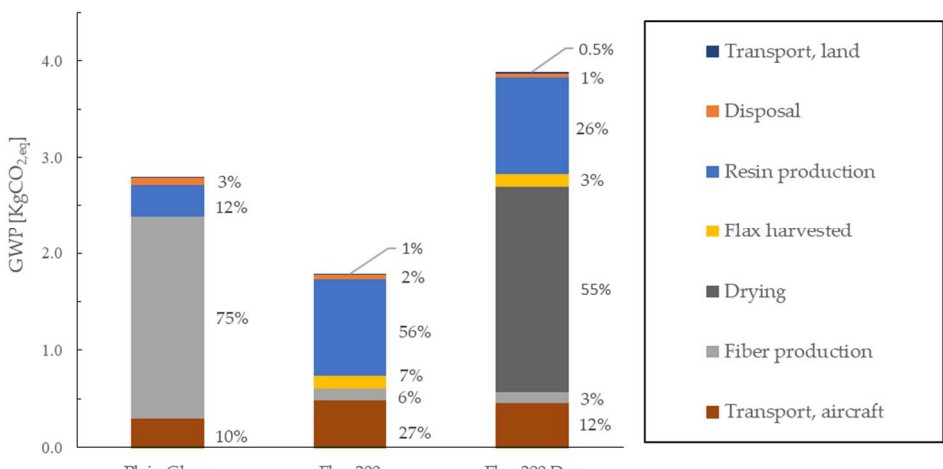

**Figure 7.** Climate impact for GWP to produce lab panels with baseline and flax alternatives. The percentages next to the bars are the relative contributions of the process steps. Those of electricity for incineration and recycling are not shown (as they are <0.5%)).

**Table 5.** Experimental and FEA data for the composite systems.

| Specimens | Flexural Strength (MPa) | FEA Flexural Strength (MPa) | % Error between FEA and Experiment |
|---|---|---|---|
| Glass Plain Weave | 115.77 ± 5.01 | 111.50 | −3.70% |
| Twill Glass | 380.00 ± 40.61 | 148.00 | −61.10% |
| Flax 200 | 178.94 ± 6.54 | 158.00 | −11.70% |
| Flax 200 Dry | 245.01 ± 11.71 | 132.00 | −46.20% |
| Flax 300 | 179.00 ± 15.78 | 190.00 | 6.20% |
| Flax 300 Dry | 190.18 ± 16.43 | 191.00 | 0.50% |
| Specimens | Flexural Stiffness (GPa) | FEA Flexural Stiffness (GPa) | % Error between FEA and Experiment |
| Glass Plain Weave | 4.50 ± 0.19 | 2.31 | −48% |
| Twill Glass | 12.60 ± 0.26 | 2.29 | −81% |
| Flax 300 | 2.70 ± 0.19 | 1.07 | −60% |
| Flax 300 Dry | 4.50 ± 0.32 | 1.10 | −75% |
| Flax 200 | 2.30 ± 0.18 | 1.06 | −53% |
| Flax 200 Dry | 2.80 ± 0.23 | 1.05 | −62% |

The chart in Figure 7 shows that the natural fiber impact production (harvest and retting) only accounts for 6 to 13% in the flax scenarios. However, flax coupons have a higher impact in terms of resin production associated with their fabrication, possibly attributed to the greater amount of resin used in fabricating the coupons. Flax composites in this work require at least 2.5 times more resin than the glass variants, resulting in a much thicker composite overall (as seen in Table 3).

The effect of the HTP indicator for glass fiber production is similar to that of flax production. The impact of glass fiber production on human toxicity is primarily occurring due to cadmium (Cd), antimony (Sb), and hydrogen fluoride emissions (HF). The POCP of flax production is slightly lower than the POCP of glass fiber production (by 4%). This is because more resin was required to fabricate flax-reinforced composites, resulting in significantly heavier panels, which contributed to increasing the impact of this category.

Accordingly, the EP of flax production is 45% higher than the EP of glass fiber production, owing to the greater distance of raw materials shipment of flax. Furthermore, there is a slight difference in AP between plain glass and dry flax production. However, it should be noted that the AP occurring from wet flax production is significantly lower than the AP of glass fiber production and much lower than the dry flax variant. Dry flax panels have a significantly higher impact than the other panels in all categories as their production requires the use of a furnace for 12 h, which in turn increases the electricity use by orders of magnitude as opposed to the non-dried flax and glass fabrics.

A contribution analysis for all environmental impact indicators revealed that the various process phases do not contribute equally to the overall climate impact (see Figure 8). Eutrophication is the category that seems to decrease flax composites' performance, and flax cultivation is associated with emissions to soil and water. As a result, the composite containing flax fibers has a significantly greater impact on eutrophication than the composite containing synthetic fibers as e-glass. Figure 8 shows a comparison of contribution impacts between flax and glass variants.

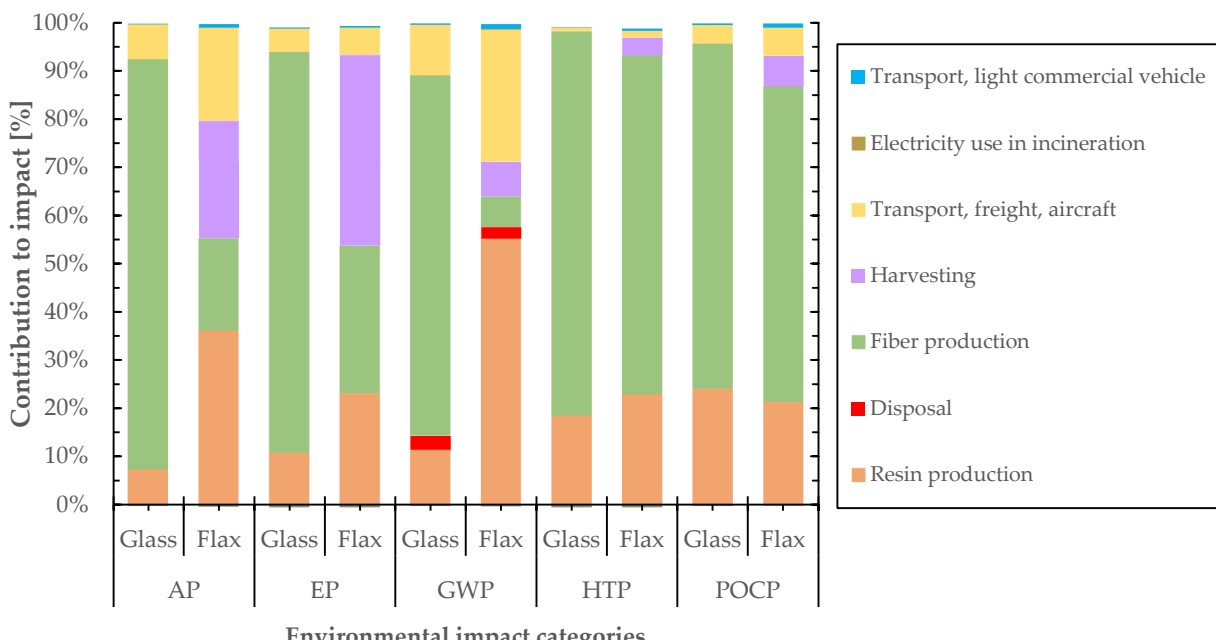

**Figure 8.** Contribution analysis for the impact categories for the plain weave glass and flax 200 wet variants.

In the case of EP, the contributions of natural fiber harvesting and retting total $4.53 \cdot 10^{-3}$ kg $PO_{4,eq}$ which is approximately 22% higher than glass production in this category. Generally, all glass scenarios have lower EP impacts as glass fiberization does not produce fertilizers or biomass formation, and synthetic fiber production does not use agricultural resources or toxic pesticides. The POCP depends on the fuel consumption in crop production and retting; therefore, higher values are observed for flax production rather than glass fiber.

According to a contribution analysis for the environmental impact categories considered in this study, human toxicity and acidification contribute the most to the environmental impact of the glass fiber-reinforced composites. Moreover, eutrophication and photochemical oxidation were the LCA categories for the wet flax variants with the highest contribution to the climate impact per kg. The main contributors to this impact are primarily flax production-associated upstream activities.

Compared to the emissions involved with glass fiber production, collecting and turning flax into mats is a process that typically consumes less energy. However, the transportation profile for flax scenarios represents the main energy sinks and carbon emitters.

Relative Indicator Results

The chart in Figure 9 depicts the relative indicator results of the various project variables. The maximum result for each indicator is set to 100%, and the outcomes of the other alternatives are presented in relation to this result. This graph shows the results for FU production per composite panel.

The greatest contribution to the overall life cycle impacts of the panels arose in different categories per panel. The addition of natural fibers acts as a complementary effect and increases its environmental performance, given that the reinforcements derive from annually renewable resources. This is mainly because of its non-damaging primary raw material production, non-fossil energy-demanding fiber.

A literature review adduces that the drying procedure effectively reduces surface moisture content and improves the dried coupons' mechanical performance [1,20,26]; however, this alone contributes significantly to the increase in all impact categories under study. It must be considered that drying fabrics can take up to 12–24 h for moisture to be effectively removed. However, a considerable amount of energy is spent if we account

for just one fabric to be placed in the oven, which is why this category's impact is so high. If we were considering a batch of panels (thus filling up the oven with a more oversized fabric mat), the outcome would have been less energy demanding per coupon panel.

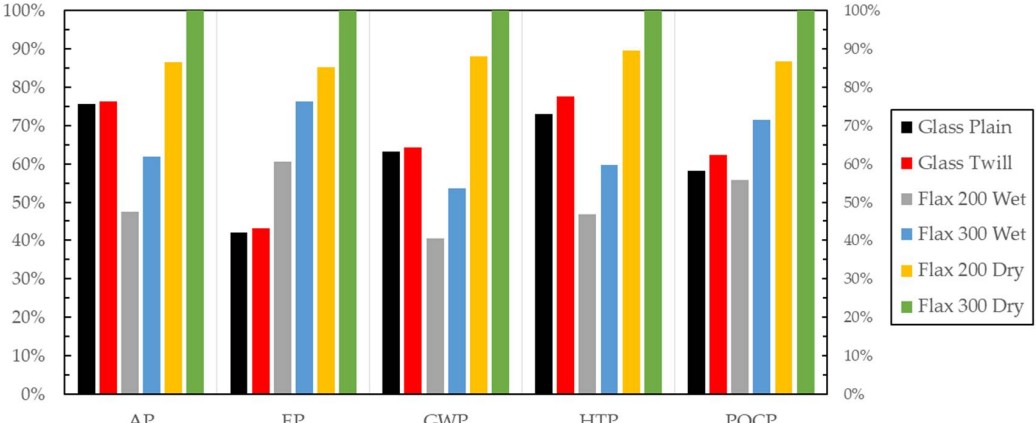

**Figure 9.** LCA relative indicator results of respective panel variants.

In addition, melting raw materials to produce glass fibers generate emissions of particulates, nitrogen oxides ($NO_x$), and sulfur oxides ($SO_x$). These particulates are generated from fuel combustion and dissociation of raw materials and are primary air pollutants. The software tools $NO_x$ and $SO_x$ are often lower when considering the flax fabrics. However, shipping flax fabrics from overseas (from Europe to Asia specifically) increases all impact categories until fibers reach the manufacturing facility. And the following analysis shows the same composite setting by swapping the fibers' location as if these were supplied from an Asian supplier. The authors did not identify any producer that could potentially supply fabrics of the same quantity and quality; however, we presented the outcome of such a possible scenario.

Figure 10 compares all relative indicator results by environmental impact category, including a scenario in which the flax fiber is imported by a Chinese (CN) supplier to improve its environmental performance further. Table 6 shows the LCA results of the compared systems. Each selected impact category is displayed in the rows, and the variants of two compared composite systems are in the columns. The impact indicator is the unit of the LCA category as defined by the methods used. It is evident by the chart tabs that introducing a fiber that is closer to the manufacturing facilities lowers all categories investigated. It is observed a decrease in energy and $CO_2$ emissions. Additionally, this rendered the CN scenario the most advantageous over the synthetic glass variant besides the EP category (in Table 7). The contributions of the processes to eutrophication are largely affected by the change to a cleaner energy background system and for the reasons regarding fiberization explained earlier.

**Table 6.** Comparison of the total impacts to produce one composite panel per variant.

| Impact Categories | GWP | AP | EP | HTP | POCP |
|---|---|---|---|---|---|
| | [kg $CO_{2,eq}$] | [kg $SO_{2,eq}$] | [kg $PO_{4,eq}$] | [kg 1,4-$DB_{eq}$] | [kg $C_2H_{4,eq}$] |
| Glass Plain Weave | 2.79 | $1.55 \times 10^{-2}$ | $4.39 \times 10^{-3}$ | 2.91 | $6.54 \times 10^{-4}$ |
| Glass Twill Weave | 2.84 | $1.56 \times 10^{-2}$ | $4.51 \times 10^{-3}$ | 3.09 | $7.01 \times 10^{-4}$ |
| Flax 200 Wet | 1.79 | $9.76 \times 10^{-3}$ | $6.37 \times 10^{-3}$ | 1.88 | $6.28 \times 10^{-4}$ |
| Flax 200 Dry | 3.88 | $1.77 \times 10^{-2}$ | $8.92 \times 10^{-3}$ | 3.57 | $9.74 \times 10^{-4}$ |
| Flax 300 Wet | 2.37 | $1.27 \times 10^{-2}$ | $8.00 \times 10^{-3}$ | 2.39 | $8.04 \times 10^{-4}$ |
| Flax 300 Dry | 4.41 | $2.05 \times 10^{-2}$ | $1.05 \times 10^{-2}$ | 3.99 | $1.12 \times 10^{-3}$ |

**Table 7.** Comparison of the total impacts of producing one flax panel with resources closer to local production.

| Impact Categories | GWP | AP | EP | HTP | POCP |
|---|---|---|---|---|---|
| | [kg $CO_{2,eq}$] | [kg $SO_{2,eq}$] | [kg $PO_{4,eq}$] | [kg $1,4\text{-}DB_{eq}$] | [kg $C_2H_{4,eq}$] |
| Flax 200 Wet | 1.79 | $9.76 \times 10^{-3}$ | $6.37 \times 10^{-3}$ | 1.88 | $6.28 \times 10^{-4}$ |
| Flax 200 Wet Cn | 1.47 | $8.55 \times 10^{-3}$ | $6.13 \times 10^{-3}$ | 1.85 | $6.00 \times 10^{-4}$ |

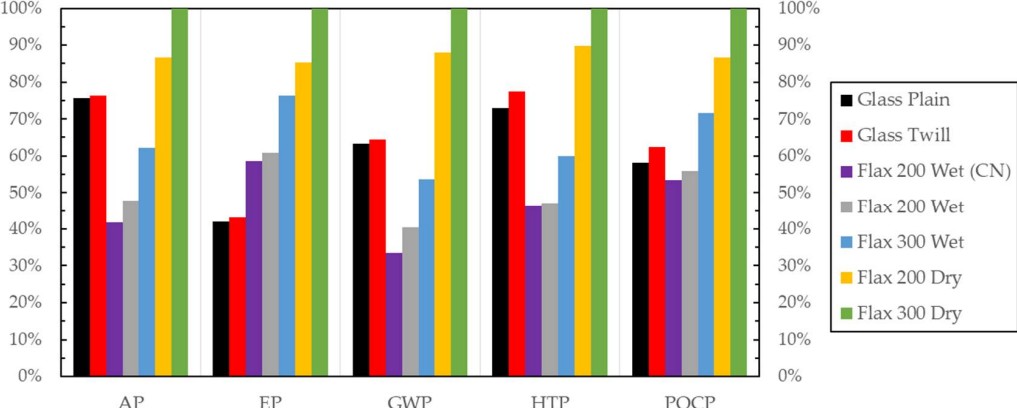

**Figure 10.** LCA relative indicator results of respective panel variants.

## 4. Discussion

The need to substitute synthetic compounds with renewable substances composites is becoming imperative. Not only as a recent trend but also a necessity for a more sustainable society in terms of materials used. Green composites require improvements to compete with regular composites, mainly when mechanical performance is a critical driver. As such, general conclusions point towards the enhanced structural performance of glass fiber composites. Inverse results occurred regarding environmental performance as assessed by software packages granting better performance to plant-based composites.

Generally, good agreement between the predicted and experimental values was observed; however, the model underestimated several properties. Other factors contribute to the variation of the simulation, including the volume of fibers, the density of composite preforms, and the adaptability of/to the particular manufacturing process [25]. Fiber treatments can be employed for natural fibers to enhance their mechanical performance, with an associated decrease in the environmental performance of the composites [12]. Nonetheless, most studies suggest the potential of natural fibers, particularly for lightweight construction, due to their reduced density (about 1.5 g/cm$^3$ versus 2.5 g/cm$^3$ for glass) and superior mechanical properties [16–18].

Examples presented here show genuine prospects for this approach in product and process design, which may result in less CO$_2$-intensive solutions and decrease the use of fossil resources in composites production. Furthermore, the environmental benefits of green composites may be negated if they have a shorter operating life than their conventional counterparts [16]. Synthetics outperform naturals in terms of eutrophication impact and, in some cases, have a better POCP profile. Using fertilizers to produce natural fibers results in higher nitrate and phosphate emissions, which increases the amount of eutrophication in the local environment. As a result, the overall environmental impact improves at the expense of the local environment and water quality deterioration [36]. The natural fibers studied here are subject to challenges regarding their geographical availability, which is the main contributor by an order of magnitude higher than the POCP.

The environmental performance of flax panels is comparable to that of glass in the POCP category, and the production might be decreased by minimizing the supply chain distance. This environmentally damaging variable can be further reduced by bringing the fabrics from a manufacturer closer to the lab. Thus, compared to the original flax scenario

(in Figure 10), in which the fabric mats could be sent from China, the overall impact of natural fiber may be decreased by 3% in energy and 5% in $CO_2$ emissions. Analogous results have been reported in earlier studies when importing fabrics from suppliers closer to manufacturing facilities [20].

The use of greener materials with embodied energy linked with their manufacturing can significantly reduce environmental consequences in GWP, AP, and HTP categories. Furthermore, as demonstrated earlier, EP and PCOP impacts can be further reduced by utilizing constituents derived from local resources. The location of the natural fiber plantation was proven to be extremely important and is one of the reasons why flax was not considered the most environmentally friendly fiber in some categories. This shows that to maximize environmental advantages, it is not enough to substitute materials without considering the provenance of the resources.

When it comes to enhancing the environmental performance of dry fabrics, the three key areas are eutrophication, acidification, and global warming. Therefore, these categories must be considered and optimized to make green composite systems sustainable. If so, using materials of a greener profile with embodied energy associated with their production can significantly reduce the overall environmental impacts. Furthermore, if the matrix resin is biodegradable, these fully green composites can be recycled or composted [19,43,44].

Although biobased composites offer significant advantages, their reuse and recycling are problematic. The most frequent procedures are direct disposal in landfills or incineration, which involve high costs and technological challenges and have environmental consequences [36]. A recycling scenario was considered in this work; however, future studies may well delve into how these green composites can be optimized for optimal recycling and downcycling rates to achieve cost-effectiveness.

## 5. Conclusions

The primary goal of this study was to assess the mechanical and environmental performance of prospective composite panels that could be applied in prospective structural parts. Glass and flax/epoxy composite sheets were manufactured, and an FEA model was developed to simulate the flexural performance of these composite specimens reliably.

With regard to the sustainability of the reinforcements, we conducted a qualitative LCA of five environmental impact classification factors: global warming, acidification, eutrophication, ozone depletion, and human toxicity. The analysis considers flax has improved environmental credentials in production and shows the clear advantages of swapping from synthetic fiber to natural fiber in the scope of our analysis. The climate impact of glass fiber production scenarios was 2.79 and 2.84 kg $CO_{2,eq}$/kg, roughly 20% higher than the wet flax fiber counterparts. When considering dried flax composites, limitations outweigh the benefits over synthetic fabrics as more $CO_{2,eq}$/kg is generated (30% higher) due to the high energy associated when oven-drying the fiber mats.

FEA and LCA have proved to be efficient methods to be used collaboratively and assess materials suitable for future green applications. An important question is whether these composites can be developed to attain equivalent ecological and mechanical performance as their predecessors while at the same time being cost-effective.

**Author Contributions:** A.S. conceived and guided the affiliated project and contributed to environmental assessment. G.K. analyzed the experimental data and interpretation of the environmental output. M.O. fabricated and tested some of the specimens under study. The corresponding author has drafted the outcome of this work, and the co-authors made a critical review and a substantial revision of the submitted manuscript overall. All authors have read and agreed to the published version of the manuscript.

**Funding:** This work was funded under the research project PIE-SGP-AN-2019-01 Rethinking design spaces for composite structures from the Singapore University of Technology and Design.

**Acknowledgments:** The authors would like to acknowledge the work of Nurul Zarifah Binte Ibrahim on the fabrication of some of the specimens tested.

**Conflicts of Interest:** The authors declare no conflict of interest.

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
