# Peer review of "Comparison of Structural Performance and Environmental Impact of Epoxy Composites Modified by Glass and Flax Fabrics"

_jcs, doi:10.3390/jcs6100284_

Round 1

Reviewer 1 Report (Previous Reviewer 1)

The authors satisfied the comments and suggestions provided and the revised version of manuscript has a quality for the publication. The disagreement between FEA and experimental results are still an issue but the explanation of disagreement is good enough. LCA part of the literature still can be improved, authors may look at different publications or reviews to enlarge the literature such as  Sustainability 2021, 13(3), 1160; https://doi.org/10.3390/su13031160 .

Author Response

Additional studies have been cited to this revision. All edits are in blue fonts

Reviewer 2 Report (Previous Reviewer 2)

Authors provided a well-thought explanation to the reviewer's comments;  however, please add infomation related to the comment:

What is the microstructure of the obtained composites? SEM technique could provide useful information.

Author Response

We did not perform SEM on the composite panels but on the fabrics themselves to obtain properties for the FEM model. We have added the following in section 2.2  

“Scanning electron microscopy (SEM) was used to examine the morphology of the fabrics” 

Round 2

Reviewer 1 Report (Previous Reviewer 1)

The authors satisfied the comments and provided a high level of revised version of the manuscript. However, I still believe the difference between FEA and experimental results could have been investigated more deeply. But If authors believe that this is their best of knowledge and best of their presentation, it is acceptable. I can still spot confusing sentences, please go through the text and try to make clear for the reader. Such as line 64-67 "Moreover, a direct comparison between synthetic and natural fiber composites is sometimes hard to perform because of fundamental geometry differences (volume of fibers, the density of performs, adaptability of/to the particular manufacturing process, etc.)." I did not understand what "density of performs" means. Also, please have a look at Materials 2020, 13(9), 2129; https://doi.org/10.3390/ma13092129, for examples of direct comparisons via the modified rule of mixtures. After these minor corrections, the manuscript can be acceptable.

Author Response

The sentence has been revised to include the study mentioned and corrected mistyped word from "perform" to "preform". It reads now as follows: 

Moreover, while a direct comparison between synthetic and natural fiber composites is possible (via a modified rule of mixtures) [24], it is sometimes hard to conduct due to fundamental geometry differences (volume of fibers, the density of preforms, adaptability of/to the particular manufacturing process, etc.) [25].

[24]  A. Kandemir, T. R. Pozegic, I. Hamerton, S. J. Eichhorn, and M. L. Longana, "Characterisation of Natural Fibres for Sustainable Discontinuous Fibre Composite Materials," Materials, vol. 13, no. 9, p. 2129, 2020.

[25]  G. Koronis, A. Silva, and M. Fontul, "Green composites for an electric vehicle body: a review of adequate materials’ combination," presented at the 16th International Conference on Composite Structures, Porto, 2011.

Reviewer 2 Report (Previous Reviewer 2)

Authors properly addressed the reviewer's comments, and the revised manuscript can be published.

Author Response

Thank you for the constructive feedback!

Round 3

Reviewer 1 Report (Previous Reviewer 1)

All comments are satisfied.

This manuscript is a resubmission of an earlier submission. The following is a list of the peer review reports and author responses from that submission.

Round 1

Reviewer 1 Report

Authors presented the experimental and modelling of 3-point bending results for glass and natural fibre reinforced epoxy composites with further LCA analysis on them. The manuscript sadly prepared poorly, the English is not very understandable, and most importantly, their modelling results are questionable. There are serious unsupported statements either needs references or characterisation proofs. Because of that, I suggest rejection for this manuscript. 

-Affilations of the authors are missing.

-Abstract has some sentences which are difficult to understand and not meaningful. (lines 10-12, lines 18-20)

-Abstract does not actually reflect the work, must be re-written and must be more clear about what has been investigated and what the results are.

-Keyword fabrication is too general, can be changed.

-Introduction has some statements that needs references (such as even first sentence, lines 41-42, 42-43) and literature review of the paper must be improved. Moreover, referencing is not in order and language must be improved since it is quite informal.

-"Flexural tests are frequently used for quality control in composites, despite being rarely used for design." Please avoid statements like this, or reference them. 

-Line 96: what is procedure A? why do the authors not being clear and informative to the reader?

-"Table 1 shows how ASTM-recommended dimensions were used" Table 1 has a caption of "Parameters used in the finite-element model"-> The method section is confusing and not clear.

-Section 2. The ASTM standard mentioned needs to be referenced.

-"The stacking sequence applied is [(0/90)2, (45/45) 2, (0/90) 2] produced by resin infusion 106 in the university lab." the university lab? How is this important or worth to mention the reader? Rather than these information, please give details that can be meaningful for representative production.

-Please remove "belove" in the sentence - line 108.

-What or where is Section 2.2 before Sec.2.2.Xs? It is missing in the manuscript.

-Table 3 does not show Panels properties

-Line239 - ISO LCA standards? please reference them!

-The ..... could be useful for this work to mention and for readers, please have a look.

-Figure 4 is not clear and helpful for reader please add schematic version of it next to the picture.

-Is figure 5 necessary? if yes please combine all of the 3 point bending test related model, schematic, actual test in a figure.

-Line263 "In this section, we -are analyzing- the experimental data obtained from the -destructive testing phase-." The use of English is not appropirate, please use proof-reading services.

-Unfortenautly, the discussion about difference between FEA analysis and experimental results are not valid. I would expect that stifness should be independent of size and the models should predict it very closely. Experimental data shows bigger values compared to FEA data, therefore the void content is not the reason for the explanation of the difference since voids will decrease the mechanical properties. Moreover, the model does not predict the increasing trend from Flax 200 to Flax 200 Dry and from Glass Plain to Twill Glass. FEA results in this work looks like invalid. Therefore, lines 426-427 is not true.

-The discussion part does not include any literature references. I believe that there are other studies that can be used to improve the discussion especially for the LCA part of the study.

-Conclusion - Last paragraph of the conclusion needs to change. And last sentence of the conclusion "Future studies may well delve into 479 how these green composites can be optimized for optimal recycling and downcycling 480 rates to achieve that partially." is not a conclusion or an outcome from this study. I would suggest avoiding vague sentences from the manuscript. 

Reviewer 2 Report

In this work advantages and disadvantages of using natural and synthetic fibers in epoxy composites have been discussed in relation to the structural performance of fibers and their environmental impact. Mechanical behavior of fiber-reinforced composites was investigated by a three-point bending test and the life cycle impact analyzed. As composites containing lignocellulosic fibers are becoming an important part of „green” materials, this work follows the current trends. Several issues need however to be addressed in more depth:

- please provide the affiliations of the authors,

- The title is too general – the materials invesigated are epoxy composites modified by glass fiber fabrics and flax textiles,

- „Introduction” should be completed/expanded with information on previous works dealing with epoxy composites modified by glass fibers / flax,

- More information on the sizes and geometry of glass fiber fabrics and flax textiles used in the course of this work should be provided,

- What is the microstructure of the obtained composites, e.g. by SEM technique,

- Authors used the CML-IA baseline impact technique (University of Leiden) – please provide criteria foe selection of a given flow and its characteristics,

- what about the recycling possibilities of the fabricated epoxy composites?